# Interactive Effects of Copper-Doped Urological Implants with Tissue in the Urinary Tract for the Inhibition of Cell Adhesion and Encrustation in the Animal Model Rat

**DOI:** 10.3390/polym14163324

**Published:** 2022-08-16

**Authors:** Wolfgang Kram, Henrike Rebl, Julia E. de la Cruz, Antonia Haag, Jürgen Renner, Thomas Epting, Armin Springer, Federico Soria, Marion Wienecke, Oliver W. Hakenberg

**Affiliations:** 1Department of Urology, Rostock University Medical Center, Schillingallee 35, 18057 Rostock, Germany; 2Department of Cell Biology, Rostock University Medical Center, Schillingallee 69, 18057 Rostock, Germany; 3Jesús Usón Minimally Invasive Surgery Centre, Carretera N-521, Km. 41.8, 10071 Cáceres, Spain; 4Institute for Polymer- and Production Technologies e. V., Alter Holzhafen 19, 23966 Wismar, Germany; 5Institute for Clinical Chemistry and Laboratory Medicine, Medical Center, Faculty of Medicine, University of Freiburg, Hugstetterstraße 55, 79106 Freiburg, Germany; 6Electron Microscopy Center, Rostock University Medical Center, Strempelstraße 14, 18057 Rostock, Germany; 7Materion GmbH, Alter Holzhafen 15, 23966 Wismar, Germany

**Keywords:** copper-doped urinary stent, encrustation, animal model, in vitro, biofilm

## Abstract

The insertion of a ureteral stent provides acute care by restoring urine flow and alleviating urinary retention or dysfunction. The problems of encrustation, bacterial colonization and biofilm formation become increasingly important when ureteral stents are left in place for a longer period of time. One way to reduce encrustation and bacterial adherence is to modify the stent surface with a diamond-like carbon coating, in combination with copper doping. The biocompatibilities of the Elastollan^®^ base material and the a-C:H/Cu-mulitilayer coating were tested in synthetic urine. The copper content in bladder tissue was determined by atomic absorption spectroscopy and in blood and in urine by inductively coupled plasma mass spectrometry. Encrustations on the materials were analyzed by scanning electron microscopy, energy dispersive X-ray spectroscopy and Fourier transform infrared spectroscopy. A therapeutic window for copper ions of 0.5–1.0 mM was determined to kill bacteria without affecting human urothelial cells. In the rat animal model, it was found that copper release did not reach toxic concentrations in the affecting tissue of the urinary tract or in the blood. The encrustation behavior of the surfaces showed that the roughness of the amorphous carbon layer with the copper doping is probably the causal factor for the higher encrustation.

## 1. Introduction

The insertion of a ureteral stent is an effective treatment that restores ureteral urine flow and eliminates ureteral obstruction or dysfunction. In cases with chronic obstruction and poor prognosis, urinary stenting may be a long-term treatment, whereby the ureteral stent has to be replaced at intervals. The problems of encrustation, bacterial persistence and biofilm formation become increasingly important, especially when ureteral stents are left in place for longer periods of time. Excessive stent encrustation leads to stent blockage and subsequently to painful hydronephrosis, renal infection, and in the long-term, to impaired kidney function [1].

Encrustations of ureteral stents are caused by the precipitation of salts, predominantly calcium oxalate [2]. Patients with high excretion of salt-forming ions tend to deposit crystals on ureteral stents. This process may occur without a significant microbial load, but it favors the accumulation, multiplication and persistence of bacteria in the biofilm.

Uropathogenic microorganisms (usually Enterobacteriaceae) are either introduced into the bladder during catheter insertion or migrate along a transurethral catheter. From the bladder, bacteria ascend through the ureter and especially along a ureteral stent into the kidneys [3,4]. This results in a high contamination and bacteriuria rate. The distribution of bacteria in the urinary tract occurs frequently, but in most cases, it does not lead to clinical problems if urine outflow is unobstructed and the biological function of the urinary tract is sufficient. 

In the case of biofilm formation on urological implants, persisting bacteria are protected from the natural defense mechanisms. This leads not only to more clinically relevant urinary tract infections, but also to antibiotics being less effective. In addition, bacteria in biofilms have a reduced metabolic rate, which further reduces the effectiveness of most antibiotics, and consequently promotes antibiotic resistance [5,6].

In principle, it is undisputed that the surface properties of a biomaterial in the urinary tract (e.g., coating, charge, topography, roughness, as well as virulence factors of microorganisms and adhesins) have an impact on the speed and extent of biofilm formation. Most ureteral stents used are made of polymer blends with anti-adhesive properties [7]. Current biocompatible polymers, e.g., Elastollan^®^ 1185 A 10 FC (BASF Polyurethanes GmBH, Mainz, Germany), Styroflex^®^ 2G66 (INEOS Styrolution Group GmBH, Frankfurt a.M., Germany) and Greenflex^®^ (Versalis S.p.A, San Donato Milanese (MI), Italy) have good mechanical stability and flexibility with anti-adhesive properties and are suitable for the development of thin-walled stents with good urine drainage [8]. Additives to the base materials are positive X-ray contrast agents, which affect the intended properties of the stents. Additives of 25–30% barium sulphate, a biocompatible salt with high electron density, are used.

One way to reduce the rate of encrustation and bacterial adhesion is to modify stent surfaces with different materials. Known for this purpose in the urinary tract are coatings with covalently bound heparin, polytetrafluoroethylene (PTFE), hydrogels, plasma deposited diamond-like carbon (DLC) and urease inhibitors [9]. Common strategies are the formation of a film or release of bactericidal agents, including antibiotics, bacteriophages, nanoparticles of metal oxide, metal ions and carbon, antimicrobial peptides, zwitterionic polymers and biofilms with non-pathogenic bacteria. 

The toxic effect of metal ions on bacteria, but also on human cells, is a well-known phenomenon and has been investigated in numerous studies [10,11,12,13,14]. Heidenau et al. [15] evaluated the cell toxicity of several metal ions by a growth inhibition assay in the L929 cell line (mouse). Their results showed very strong cytotoxicity for Ag, Zn and Hg ions at low concentrations, while the tissue cells tolerated relatively high concentrations of Cu and Al ions. There is a controversial debate about a possible inactivation of silver-mediated antibacterial activity in physiological fluids and about the low biocompatibility of silver, which is determined by the low threshold concentration for cytotoxic effects [16]. Li et al. [17] showed that silver nanoparticles had the potential to induce embryonic cytotoxicity.

In the present in vivo studies, the interaction of amorphous hydrogenated carbon thin film coatings (a-C:H), also called diamond-like carbon (DLC), in combination with nanoscale copper-multilayers (a-C:H/Cu-multilayer), with the tissue of a rat urinary bladder and the encrustation of the implants were investigated. A temporally and quantitatively defined release of the antibacterial substance copper (copper ions) should reduce the development of bacterial infections.

## 2. Materials and Methods

### 2.1. Implant Preparation

Different approaches for a copper-containing diamond-like carbon coating have been compiled for this work.

In particulary, the coating should have a high starting release of antibacterial copper ions for a sure devitalization of any microbes and, on the other hand, it must not damage the living cells surrounding. Against this background, it was aspired to adjust the time-dependent release of copper-ions in a defined manner, and thus to adjust the biocompatibility over longer time. Therefore a prototype plasma coating process have been developed to fill a-C:H coatings with nanoscale copper multilayers in a hybrid PVD-PECVD process.

#### 2.1.1. Implants with Copper Coating Beads

Borosilicate glass beads have been coated with about 1 µm thick pure copper coatings by RF-magnetron sputtering (process parameters see Table 1 process no.2, PVD).

Group 1 (glass)

The group included 2 mm glass beads, borosilicate glass 3.3, technical data sheet 0570, Hilgenberg GmbH, Malsfeld, Germany. 

Group 2 (Cu)

The group included 2 mm glass beads, borosilicate glass 3.3 with copper coating, Hilgenberg GmbH, Malsfeld, Germanyand coating from Materion GmbH, Wismar, Germany.

Group 3 (Cu+)

The group included 2 mm copper beads, grade 200, product code CU00-SP-000130, Goodfellow GmbH, Hamburg, Germany.

#### 2.1.2. Impants with a-C:H/Cu-Mulitilayer Coating

Elastollan tube samples with a diameter of 2 mm and a thickness of 2 mm have been coated with a-C:H/Cu-multilayers as shown Figure 1 (Elastollan^®^ 1185 A 10 FC, BASF Polyurethanes GmBH, Mainz, Germany).

Group A: Elastollan^®^, provided with the X-ray contrast agent barium sulphate, Institute for Polymer Technologies e.V., Wismar, Germany 2.Group B: Elastollan^®^, provided with the X-ray contrast agent barium sulphatesulphate and a-C:H/Cu-multilayer coating in combination with copper doping, Materion GmbH, Wismar, Germany.

Plasma enhanced chemical vapor deposition (PECVD) of diamond-like carbon and magnetron sputtering of copper (physical vapor deposition, PVD) were performed, alternating to create multilayers with time-dependent copper release. The single copper layers have a thickness of about 200 nm, separated by the a:C:H interlayers. The top layer is pure copper, with a thickness of about 1 µm and cauliflower-like morphology.

Table 1 summarizes the process parameters of the hybrid PECVD/PVD process. For the long time release, 4 times process 1 and 4 times process 2 have been performed alternating. For the high initial Cu-release at least 4 times process 2 was performed.

### 2.2. In Vitro Studies

#### 2.2.1. Biocompatibility of the Materials

For urothelial cells, the biocompatibility of the samples was studied using human non-tumorigenic urothelial HUC-1 cells (ATCC, CRL-9520, LGC Standards GmbH, Wesel, Germany). Cells were cultivated in Dulbecco’s Modified Eagle Medium (DMEM, Thermo Fisher Scientific, Waltham, MA, USA) with 10% fetal calf serum (FCS, PAN Biotech GmbH, Aidenbach, Germany) and 1% antibiotic-antimycotic solution (Thermo Fisher Scientific) at 37 °C in a humidified atmosphere with 5% CO_2_. The MTS assay (CellTiter 96^®^ Aqueous One Solution Cell Proliferation Assay, Promega, Madison, WI, USA) was performed to assess the biocompatibility of the materials. The principle of the test includes an enzymatic cleavage of the methyltetrazolium salt (3-(4,5-dimethylthiazol-2-yl)-5-(3-carboxymethoxyphenyl)-2-(4-sulfophenyl)-2H-tetrazolium, inner salt (MTS)) by metabolically active cells into a formazan product. The amount of the formazan product is directly proportional to the number of living cells. Then, 4 × 10^4^ cells were seeded into 12-well cell culture plates (TCPS, Greiner, Kremsmünster, Austria) and the polymer samples (6 mm diameter) were added via ThinCert inserts (8 µm pore size, Greiner, Kremsmünster, Austria) at a volume of 2 mL. For determination of cell sensitivity towards copper, the cells were incubated for 24 h with medium containing different amounts of CuCl_2_. In both cases, the medium was exchanged after 24 h and 100 µL of the MTS solution was added to each well and incubated for 2 h. Subsequently, 100 µL of the supernatant was transferred in triplicate to a 96-well plate, and spectrophotometric absorbance was analyzed at 490 nm using an ELISA reader (Anthos 2010, Anthos Labtec Instruments, Wals, Austria).

#### 2.2.2. Anti-Bacterial Effect of Copper

For the study of the anti-bacterial effect of the materials, the bacterial strain Escherichia coli HB 101 was used. These uropathogenic bacteria are responsible for a substantial number of urinary tract infections. The bacteria were cultured in soybean-casein digest medium. To simulate the in vivo conditions, the material tests were performed in synthetic urine. For the experiment, 5 × 10^5^ CFU/mL were seeded in 1 mL synthetic urine in a 24-well plate and incubated with CuCl_2_ at various concentrations [8,18]. The covered plate, sealed with parafilm, was placed in a humidity chamber and then incubated for 24 h in a shaker incubator at 37 °C and 100 rpm. After incubation, a sample was taken from each well and a dilution series was prepared. Then, 30 µL of the dilution steps and the undiluted sample were added to a section of the prepared agar plate. The agar plates were cultured in an incubator at 37 °C for approximately 18 h, after which the analysis was performed. The plates were scanned, colonies were counted and CFU values/mL were calculated from the dilution levels with countable bacterial load.

### 2.3. Animals and Housing Conditions

Animal experiments in rats were performed in accordance with the national animal protection guidelines and the accepted principles of the welfare of animals used in science and were approved by the regulatory authority (AZ 7221.3-1-020-17 and AZ 7221.3-1-011/14-3). Throughout the study, all animals had free access to water and standard rat chow. Male Sprague-Dawley rats (8 weeks of age, approximately 300 g of weight) were purchased from Charles River Laboratories Research Animal Models (Charles River Research Models and Services Germany GmbH, Sulzfeld, Germany). After an acclimatization period of five days, the animals were randomly allocated to the experimental groups. 

#### 2.3.1. Interaction of Copper-Doped Implants with the Tissues of the Urinary Tract

In the rat animal study AZ 7221.3-1-020-17, the influence of copper-doped implants on the damaged urothelium of the urinary bladder and the effect on the entire bladder tissue was investigated.

Group 1 (glass) included borosilicate glass, 2.0 mm beads without coating, *n* = 18, Materion GmbH, Wismar, Germany. 

Group 2 (Cu) included borosilicate glass, 2.0 mm beads with copper coating, 2.85 µm, 0.319 mg, *n* = 21, Materion GmbH, Wismar, Germany.

Group 3 (Cu+) included 99.9% O.F.H.C copper, oxygen-free, grade 200 604-930-79, 2.0 mm solid copper beads, *n* = 17, Goodfellow GmbH, Hamburg, Germany.

Surgical access was via a very small median lower abdominal incision. The distal part of the bladder was incised to implant the specimens. The intraoperative application of 0.6 mL protamine sulphate (10 mg/mL, LEO Pharma GmbH, Neu-Isenburg, Germany) as a bolus over 30 s into the urinary bladder led to a selective ablation of the urothelial cover cells (microinjection, 33G, 0.20 × 12 mm, Mesoram^®^, PromaMedical GmbH, Munich, Germany). After 15 min of exposure, the urinary bladder was flushed with isotonic saline (NaCl 0.9%, Braun Melsungen AG, Melsungen, Germany) to neutralize the protamine sulphate injection. In all animals, urine and blood were sampled pre-operatively. Postoperatively, daily urine samples were collected and blood was taken on postoperative day 5 and on day 21. The aim was to evaluate renal function (serum creatinine), inflammatory processes (serum leukocytes), wound healing disorders (serum thrombocytes) and anemia (serum hemoglobin). Postoperative analgesia was given for 7 days (metamizole, 1 mL Novaminsulfon^®^, Zentiva Pharma GmbH, Frankfurt am Main, Germany, 500 mg/mL dissolved in 400 mL water). During the observation period, a vigilance score, body weight, food intake, mobility and appearance of the animal were monitored. In addition, urine cytology, as a simple method for the non-invasive assessment of the course of wound healing in the urinary tract, was examined. On day 21, the animals were sacrificed by an anesthetic overdose and CO_2_. The urinary tract was removed, preserved in 4% paraformaldehyde, and embedded in paraffin for histological examination. Macroscopic evaluation also included an assessment of the kidneys, adrenal glands and peritoneum. For each animal, the weight of both kidneys was determined. For the semi-quantitative evaluation of the animals’ condition, a scoring system was used [19]. 

All light microscopic examinations of copper accumulation in the urinary bladder tissues were performed with hematoxylin-eosin (HE) staining, a survey stain for morphological examinations and immunohistochemical rhodanine staining (microscope BX43 Olympus, digital camera CAM-XC30 with cellSens DIMENSION-V1.7 Count+Measure). The reagent (5-p-dimethylaminobenzylidene-rhodanine) binds to both the free copper moiety and copper-associated proteins, forming a reddish-brown color complex (Wilson’s Disease Stain, Bio Optica Milano S.p.A., Milano, Italy). A blue nuclear stain with hemalaun was used for contrast. 

The bladder tissue was analyzed using flame atomic absorption spectrometry (F-AAS, system AAnalyst400, Perkin Elmer). In order to be able to detect the low Cu concentrations, 70–100 mg of tissue, corresponding to approx. 10 mg dry weight, were required for one measurement. The lower detection limit of copper in tissue (flame atomic absorption spectrometry, F-AAS) is 8 µg/g dry tissue. The limit and reference values of the method were determined in preliminary tests. Positive tests were performed with native rat bladder tissue spiked with a copper reference solution of 100 mM copper chloride dihydrate (6.3546 µg copper/µL). 

Copper blood levels (serum) and copper levels in urine were determined by inductively coupled plasma mass spectrometry (ICP-MS).

#### 2.3.2. Encrustation of Elastollan^®^ with a-C:H/Cu-Mulitilayer Coating

A test of the results of the in vitro investigations on the encrustation tendency of the favored materials was achieved in the animal experiment AZ 7221.3-1-011/14-3 by implanting platelets with a diameter of 2 mm and a thickness of 2 mm into the urinary bladder of the rat. Elastollan^®^ 1185 A was used as the base material [8]. 

Two groups were included in the study, each with eighteen animals and an implant retention time of 21 days.

Group A included Elastollan, provided with the X-ray contrast agent barium sulphate (reference group), *n* = 18 (Institute for Polymer Technologies e.V., Wismar, Germany).

Group B included Elastollan, provided with the X-ray contrast agent barium sulphate and a-C:H/Cu-mulitilayer coating, *n* = 18, (Materion GmbH, Wismar, Germany). 

To induce encrustations (calcium oxalate), ethylene glycol (5 mL in 500 mL drinking water, C_2_H_6_O_2_, 99.5%, Carl Roth GmbH + Co KG, Karlsruhe, Germany), a precursor of oxalates in a non-toxic dose, was added to the drinking water of the rats. Within a few days, urinary oxalate crystals were formed with minimal kidney damage. Blood and urine samples were collected from all animals intraoperatively, then urine daily postoperatively and blood samples on postoperative days 5 and 21. The monitoring and follow-up of the animals, as well as the preparation of the tissue samples, was carried out according to the protocol (see Section 2.3.1). The encrustations of the implants were analyzed by polarized light microscopy (BX43 System Microscope, Olympus Europa Holding GmbH, Hamburg, Germany), Fourier transform infrared spectrometry (ALPHA FTIR Platinum ATR and Software OPUS^TM^ Version 7.5, reference library IR and SEARCH, Bruker Optik GmbH, Ettlingen, Germany) and electron microscopic methods (FE-SEM, Merlin VP compact, Zeiss, Germany) with an EDX detector XFlash 6/30 (Bruker, Bremen, Germany).

### 2.4. Statistics

Statistical analyses were performed with IBM^®^ SPSS^®^ Statistics version 28.0.1 software (San Diego, CA, USA). Groups without normal distribution were characterized using the Kruskall–Wallis test, followed by post hoc Dunn–Bonferroni test and Cohen’s effect size determination. Statistical comparison of the two dependent groups was performed using the paired t-test and effect size determination. Numerical data were expressed as mean ± standard deviation (SD) and median; a probability value of *p* < 0.05 was considered significant.

## 3. Results and Discussion

Studies have shown that the indwelling time of a urinary tract stent is the most important risk factor for encrustation [20,21]. Elwood et al. [22] observed that conditioning films contain calcium-binding proteins and can serve as nuclei for further crystal growth and encrustation of the ureteral stent. A study by Pak et al. [23] examined 1392 ureteral stents and found that calcium oxalate was the most common component of urinary stones. In addition, further studies showed that the composition of urinary stones correlated with stent encrustation in more than 70%, with increased levels of calcium phosphate (brushite) and magnesium ammonium phosphate (struvite) [24]. 

Stent encrustation and the formation of infection-induced urinary stones, as a result of urinary tract infections, proceed more rapidly than oxalate-induced encrustation and are enhanced by risk factors [25,26,27,28]. 

With in vitro and in vivo studies, we have investigated an anti-adhesive surface with antibacterial properties, in particular the interaction of copper release with the tissues in the urinary bladder of rats. Copper is an essential trace element in the human organism, integrated into metabolism and involved in a variety of endogenous enzymes responsible for tissue respiration, hematopoiesis and oxidative stress degradation in cells, among others. 

The daily copper requirement of an adult human is about 2 mg and is taken in through a normal diet [29]. After absorption, copper is mainly bound to serum albumin and transported to the liver. Excess copper accumulates as free copper and is excreted via the liver, kidneys and intestines [30]. Chronic intake of more than 10 mg/d copper can be toxic [31]. In rare cases, autosomal recessive disorders of copper metabolism occur (Wilson’s disease). In humans, toxic copper accumulation and the associated dysfunction particularly affect the liver, the central nervous system, the eyes and the kidneys. A mutation of the ATP7B gene has been identified as the cause of the disease [32]. In adults, the normal copper serum level is 0.56–1.69 mg/L. In 24-h urine, copper levels below 80 µg/L are normal and in spontaneous urine, <16 µg/L. The concentration of free copper in the serum (copper not bound to coeruloplasmin) is determined by calculation. The copper content of the liver is less than 250 µg/g dry weight [33,34]. References for copper in urinary bladder tissues are not known, so the value for the copper content of the liver, the primary copper store, was used.

### 3.1. In Vitro Studies

#### 3.1.1. Biocompatibility of the Materials

All the tested Elastollan materials w/o BaSO_4_, w/o a-C:H or a-C:H/Cu-mulitilayer showed very good biocompatibility (Figure 2). Neither the admixture of 25% barium sulphate into the bulk material, nor the coating with a-C:H or a-C:H/Cu-mulitilayer impaired the viability of the urothelial cells. Further tests showed that the addition of barium sulphate is safe up to a concentration of 50%.

#### 3.1.2. Anti-Bacterial Effect of Copper

In order to find out which copper concentrations affect urothelial cells or bacteria, a simulation with CuCl_2_ was performed. This showed that urothelial cells tolerated a concentration of 0.5 mM without impairment. In addition, 24 h incubation with 1 mM copper led to a reduction in viability by 20%, while a reduction of 60% occurred under the influence of 2 mM CuCl_2_. Only at a concentration of 3 mM were the cells completely eliminated. In contrast, even the small amount of 0.5 mM CuCl_2_ was sufficient to reduce the viability of the bacteria to below 1%. From a concentration of 1 mM, no living bacteria were detectable (Figure 3). This allows a therapeutic window of 0.5–1.0 mM in which the bacteria are killed without significant impairment of the urothelium.

### 3.2. Animal Studies

#### 3.2.1. Interaction of Copper-Doped Implants with the Tissue of the Urinary Tract

Urinary bladder implants can cause micturition disorders, due to the influence of the bladder volume. Unphysiologically frequent micturitions (phasic contractions without micturition) can influence copper release and provoke thickening of the bladder wall [35]. The implants used with a diameter of 2.0 mm (corresponds to a volume of 4.19 µL) are physiologically tolerable in the urinary bladder of the rat (0.4 mL bladder volume/100 g body weight). There is no risk of migration of the implant via the urethra. 

In the first step, the interaction of the copper-coated implants with the tissue in the urinary tract was assessed (Figure 4). 

Figure 4A shows a glass bead with a quantitatively defined copper coating [36]; (B) a solid copper bead (O.F.H.C copper, 2.0 mm, 99.9%, oxygen-free), with group Cu+ for maximum copper release. After seven days of indwelling time, residues of the copper coating were detected on the borosilicate glass. Copper is characterized as a labile metal. With reference to the in vitro experiments with the same implants in synthetic urine, no residual copper coating was expected [37]. Presumably, protein coatings delayed copper release.

Figure 5 shows the protamine-damaged tissue of the urinary bladder in the rat animal model rat in a HE stain. The reversible disruption of the urothelial barrier allows the assessment of the interactive effect of the implant under conditions of tissue injury (e.g., by an encrusted stent or tissue damage due to an inflammatory reaction). After 5–8 days, the covering cell layer is renewed. Toxins, such as copper ions, can enter the tissue and the bloodstream via the lesion. The thickness of the rat urothelium is very similar to that of humans. However, with a tunica muscularis/ urothelium ratio of 12.4 in humans and 0.7 in rats, functional differences in drug penetration are likely to exist [38]. 

Light microscopy analysis of the urinary bladder tissue samples with rhodanine staining showed no evidence of free copper or copper-associated proteins for any of the study groups (Figure 6). Copper accumulations are characterized by numerous inclusions, including lipids and fine-grained copper, as well as an enlarged intracrystalline space with dilatation and a cystic appearance. Histopathological evidence of copper accumulation in tissue would represent a significant toxic burden [39].

Figure 7 shows the concentrations of copper in the rat bladder tissue after sampling on day 21. The copper values in the bladder tissue for the group glass beads (mean value Cu = 1.1 µg/g dry weight) are within the detection limit of 1 µg/g dry weight. Implants of the group Cu glass bead with copper coating (0.319 mg) show no copper on the surface after 21 days. Group Cu+ implants correspond to the maximum release. All measured copper values in the tissue (mean value group Cu 6.9 µg/g dry weight, mean value group Cu+ 7.1 µg/g dry weight) are in the non-toxic range < 250 µg/g dry weight (4 µmol/g) [34]. The groups glass-Cu and glass-Cu+ differ significantly (*p* < 0.001), but no significant difference exists between the groups Cu and Cu+ (*p* > 0.05) using the Kruskal–Wallis test with the following post hoc test and determination of the effect size according to Cohen.

Figure 8 shows the median values for copper in serum (mg/L) with scatter for the implant groups glass beads (glass), coated glass beads (Cu) and copper beads (Cu+) at the time after implantation, on postoperative days 5 and 21. The copper values in the serum are in the non-toxic range of <1.69 mg/L [34]. The lower detection limit of copper in the serum and urine is 3 µg/L for the measurement by ICP-MS. The copper values are subject to individual scattering. If the body is exposed to significant stress, more copper is drawn from the stores into the blood. There are no statistically significant differences between the groups glass-Cu, glass Cu+, Cu-Cu+ or between the time points day 5 p.o./day 21 p.o., so it can be assumed that the copper release in the tissue of the urinary tract has no influence on the blood vessels.

Figure 9 shows the concentration of copper in the urine of rats (mg/L) for the implant groups glass beads (Glass), copper-coated glass beads (Cu) and for copper beads (Cu+) on day 2 and day 5 after surgery. The initial values in the Cu and Cu+ groups on day 2 after surgery were elevated (mean values day 2: Cu = 2.18 mg/L, Cu+ = 1.29 mg/L). The different release behavior of the copper-coated beads and native copper beads is probably due to the different morphology of the surface structure. 

In the group with copper-coated glass beads (Cu), the copper release was already significantly lower on postoperative day 5 and was consumed by the time of withdrawal, day 21. In the group with copper beads (Cu+), the copper release was sustained on postoperative day 5, but at a lower concentration level compared to the initial release of the group Cu on postoperative day 2 and showed no significant difference to the reference group glass. Presumably, the formation of a biofilm on the implants led to a delayed and reduced release in the Cu and Cu+ groups. 

On the other hand, the limitations of the rat animal model become clear here. Due to diurnal variations in urine composition, the rat urine was collected as spontaneous urine during the follow-ups at a time when the rats were inactive, but could not be collected from all rats at the control dates. The urine is generally highly concentrated, with an osmolality of 1500–2500 mOsm/Kg, compared to 50–1400 mOsm/Kg in humans [40,41,42]. Compared to humans (<16 µg/L), the concentration of copper in the urine of rats is elevated [34]. According to information from the breeder Charles River, depending on the breed and strain, in our case Sprague Dawley rat Crl:CD, the reference is 0.299 mg/L copper in urine (group glass). 

Urinary copper levels do not increase during urinary tract infections induced by uropathogenic Escherichia coli (UPEC) in the rodent model [43]. Here, on the other hand, patients with a urinary tract infection show an increased copper concentration in the urine. Ceruloplasmin, the primary carrier of circulating copper, correlates with the urinary copper levels. The reductive environment of urine promotes the formation of bactericidal Cu(I) ions and limits the growth of uropathogenic Escherichia coli (UPEC). Ceruloplasmin promotes the loading of iron onto apotransferrin to produce holotransferrin. Holotransferrin is internalized by urothelial cells during urinary tract infection, resulting in iron deficiency of uropathogenic Escherichia coli (UPEC). Mobilization of copper in urine during a urinary tract infection (UTI) limits bacterial growth through direct toxicity and also by hindering the bioavailability of iron, an essential nutrient [44].

Rats are particularly suitable for studies on the storage behavior of copper [45]. The Long-Evan’s Cinnamon (LEC) rat is the best-studied rodent model of hepatic Wilson’s disease and shows good agreement with the human disease [46]. Findings on the accumulation of copper in various tissues and as a diagnostic marker in urine and serum, as well as studies on metallothionein expression, favored the rat as an animal model for the present studies on the interactive effect of copper-doped implants [47,48]. 

To control the spread of infections, the toxicity of copper for contact killing has now been well researched [49]. The killing or growth inhibition of bacteria by copper ions in suspension or in culture follows different principles. 

In the extracellular environment, copper exists as a chemical equilibrium between copper(II)-ions, preferentially in the Gram-negative periplasm, and copper(I)-ions in the cytoplasm. Copper(II)-ions are biologically inert and not toxic to cells, compared to copper(I)-ions [50]. Copper(I)-ions can pass through the bacterial outer membrane into the cell by passive diffusion along the chemical gradient. Within the bacterial periplasm, the availability of copper(I)-ions can be reduced. Thus, the proportion of bioavailable copper(I)-ions is lower than the total copper content. In Gram-negative bacteria, copper homeostasis mainly takes place in the periplasm, where most of the intracellular copper is found [51]. After copper(I)-ions have entered the periplasm or have been generated by the reduction of copper(II)-ions, it can be inserted into the periplasmic cuproprotein [52]. To escape copper toxicity, especially under copper stress conditions, bacteria not only control intracellular copper homeostasis but must also repair the damage caused by excess copper. Bacterial responses to copper toxicity generally involve the secretion of excess copper and the control of copper-binding and copper-oxidizing proteins, such as metallothioneins, copper oxidases and copper storage proteins [53]. Cellular damage by copper via the formation of reactive oxygen species (ROS), thiol degradation and oxidative damage to proteins, lipids and DNA is considered one concept for the toxicity mechanism of copper [54,55].

For laboratory species, such as Escherichia coli, it has been shown that the main cause of copper toxicity is not the formation of hydroxyl radicals, but the displacement of iron by copper in essential iron-sulfur clusters, thereby inactivating important enzymes [56]. The concept of iron displacement from iron-sulfur cluster proteins does not preclude intracellular ROS generation. In fact, displacement of iron from iron-sulfur clusters leads to an increase in cytoplasmic ferrous ion, which can catalyze Fenton chemistry [57]. Consistent with this concept of copper toxicity, Park et al. showed that intracellular hydroxyl radicals are not significantly altered by the addition of copper(II)-ions to Escherichia coli. Rather, the bactericidal effect of copper(II)-ions is due to the cytotoxicity of cellularly generated Cu(I)-ions [58].

Understanding the bacterial response to copper toxicity may identify additional components of the bacterial copper response and reveal synergistic effects [59]. In particular, the combination of metal ions (iron and zinc) increases copper sensitivity [60,61].

#### 3.2.2. Encrustation of Elastollan with a-C:H/Cu-Mulitilayer Coating

Investigations of the explanted ureteral stents showed that encrustations occur primarily in the area of the urinary bladder and in the kidney [21,62]. The composition of encrustations at the proximal coil (kidney) reflected the composition of stones in patients with urolithiasis, while encrustations at the distal coil (urinary bladder) correlated with urinary tract infection and patient age [25].

Implantation of the samples was well tolerated by the rats (Figure 10). Postoperative monitoring of the rats, macroscopic examination of the urinary tract organs and microscopic examination of the urinary bladder tissue showed no abnormalities and good wound healing. Consistent experimental conditions were ensured by monitoring the animals according to the requirements of the animal experiment.

The selected concentration of barium sulphate (25%) in the base material Elastollan showed sufficient X-ray contrast for the intravesical implants (Figure 11).

After 21 days indwelling time, encrustations were detectable on all the implants from the urinary bladder of the rat (Figure 12).

Qualitative investigations were carried out by means of energy dispersive X-ray spectroscopy using the mapping method (EDX mapping). The mapping (Figure 13) showed crystals with the chemical elements calcium, phosphorus and magnesium. Presumably, these are calcium oxalate and phosphate crystals. In addition to the EDX mapping, the EDX line spectrum shows the quantitative distribution of the elements involved in encrustation.

Encrustations are mostly mixtures of substances. Depending on the excited vibrational and rotational states of the molecules, characteristic absorptions were measured using Fourier transform infrared spectrometry. For an accurate qualitative and quantitative analysis, suitable reference substances and mixtures must be available (OPUS^TM^ library, Bruker Optik GmbH, Ettlingen, Germany). FTIR analysis of the encrustations (Figure 14) confirmed that calcium oxalate (whewellite and weddellite), phosphate crystals (struvite, brushite and apatite) and protein were major constituents and correlated well with the studies on human ureteral stents [24,63].

In group B, implants with the Elastollan + barium sulphate + a-C:H/Cu-mulitilayer coating, a significantly higher tendency towards encrustation was observed (Figure 15). Mean values of the weights of the encrustations are as follows: group A, 1.117 g; group B, 1.994 g; effect size 0.4 (Hedges’ correction).

Presumably, the increased roughness of the a-C:H/Cu-mulitilayer coating is the cause of the higher encrustation (Figure 16). Diamond-like carbon (a-C:H) is a metastable state of amorphous carbon in which atoms with sp^2^ and sp^3^ hybridization coexist. The deposition process and the parameters used change the surface properties [64]. In this way, with the binding ratio of sp^3^/sp^2^ and the amount of hydrogen, the hardness of diamond-like carbon can reach up to 80% of diamond hardness. The doping of amorphous carbon layers with copper leads to an increase in surface energy and, depending on the total copper content, to an increase in roughness [65]. Various studies pursued the goal of achieving good antimicrobial properties through a-C:H coating by means of a low surface energy and a low coefficient of friction and the associated surface hydrophobicity [66,67,68,69]. Chan et al. [70] showed that higher roughness of the surfaces does not mean a reduction in antibacterial activity.

Our results showed that as the roughness of the implant surface increased, encrustation increased significantly. It should be noted that the vast majority of in vitro implant encrustation studies are based on traditional models with non-physiological properties, which resulted in poor correlation between in vitro and in vivo tests [71,72]. In addition to lithogenic substances, such as calcium oxalate, calcium phosphate, magnesium ammonium phosphate and uric acid, which are due to certain diseases or metabolic disorders, components of the urine can also modulate the process of crystal nucleation, aggregation and encrustation of the implants. These include naturally occurring substances in urine, such as proteins, glycosaminoglycans and pyrophosphate [73,74]. These substances influence their growth and aggregation by covering the crystal surface (Figure 3). Some substances are said to have both inhibiting and promoting effects (uro-mucoid and glycosamine glycans). Low-molecular substances, such as citrate, magnesium, sulphate and pyrophosphate, form soluble complexes with their binding to calcium, and thus inhibit crystal formation [75]. 

The mechanism of biofilm formation due to infection with urease-producing bacteria and crystallization models based on urease-induced pH change should be critically considered. Broomfield et al. [76] investigated the ability of urease-positive bacteria to induce encrustation on ureteral implants. They found that Proteus mirabilis, Proteus vulgaris and Providencia rettgeri have the highest urease activity and induce the highest rate of encrustation. Urease leads to the formation of ammonia by the hydrolysis of urea, which leads to an increase in urine pH. The alkaline environment leads to increased crystallization of magnesium ammonium phosphate (struvite) and calcium hydroxyapatite (apatite) [77]. Improved urological diagnostics have reduced the relative proportion of infectious stones (struvite) to 6% of all urinary stones [78,79]. There is consensus in urological treatment guidelines that infectious stones and their associated implants should be completely removed because of the risk of life-threatening infections and/or kidney damage and the high recurrence rate due to the presence of inactive bacteria protected by the biofilm [80,81,82]. 

During a short period of use (<6 weeks), typical encrustations with organic and inorganic components of urine can form on ureteral stents of infection-free patients. The primary goal for the development of ureteral stents is, therefore, an anti-adhesive surface. A sterile urine culture does not exclude colonization of the stent itself. Therefore, patients may benefit from an antibacterial ureteral stent during endourological procedures. Bacterial colonization of ureteral stents increases with an in-dwelling time of 6 weeks or more [25,63,83,84].

## 4. Conclusions

We investigated a-C:H/Cu-mulitilayer coated Elastollan^®^ implants with antibacterial properties in vitro and in vivo and the interaction of copper release with tissue in the urinary bladder of rats. All tested materials, both the base material Elastollan + barium sulphate and the a-C:H/Cu-mulitilayer coating, showed very good biocompatibility. At copper concentrations of 0.5 mM–1.0 mM, bacteria could be killed without affecting human urothelial cells.

In the rat animal model, it was found that copper release did not reach toxic concentrations in the damaged tissue of the urinary tract or in the blood (serum). With regard to our in vitro studies, it was shown that the formation of organic depositions on the implants resulted in a delayed copper release in the Cu- group and the Cu+ group. 

Urinary copper release was increased and was indicative of the difference in copper release between the copper-coated glass beads (significantly lower release after 5 days) and the copper beads during the study. The incrustation behavior of the a-C:H/Cu-mulitilayer-coated Elastollan implants was investigated in comparison to the base material Elastollan + barium sulphate. Our results showed that the increased roughness of the amorphous carbon layer with copper doping is presumably a causal factor for higher encrustation. The primary goal for the development of surfaces for ureteral stents is, therefore, an antiadhesive surface. Further studies with prototypes of ureteral stents with a-C:H/Cu-mulitilayer coatings are intended to test the suitability for laying times of up to 6 weeks in the pig animal model.

## Figures and Tables

**Figure 1 polymers-14-03324-f001:**
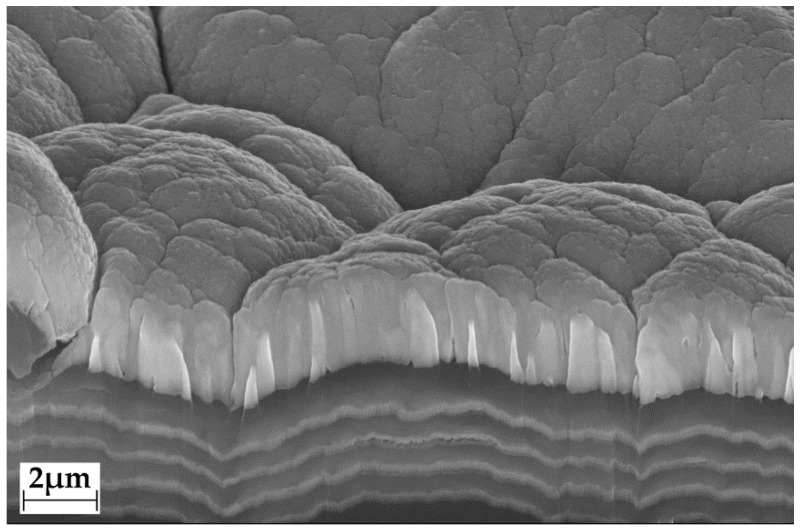
SEM picture of a a-C:H/Cu-multilayer cross section prepared by focused ion beam preparation method.

**Figure 2 polymers-14-03324-f002:**
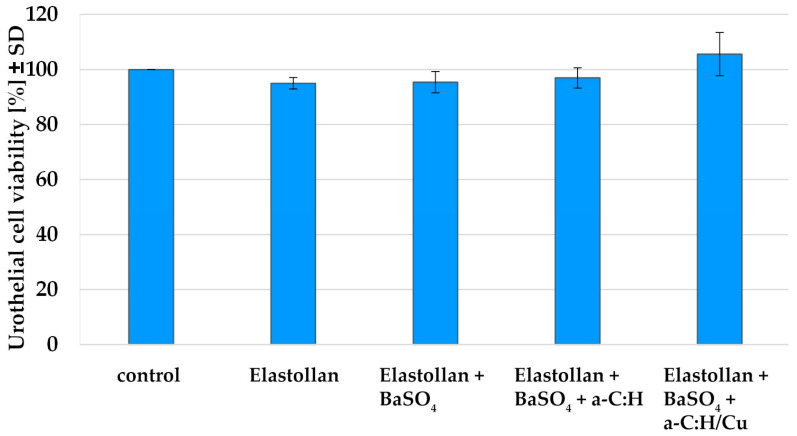
Cell viability after 24 h incubation with the different materials +/− coating. Regardless of the addition of barium sulphate, or the coating with a-C:H or a-C:H/Cu-mulitilayer, the excellent biocompatibility of the material is maintained (*n* = 3).

**Figure 3 polymers-14-03324-f003:**
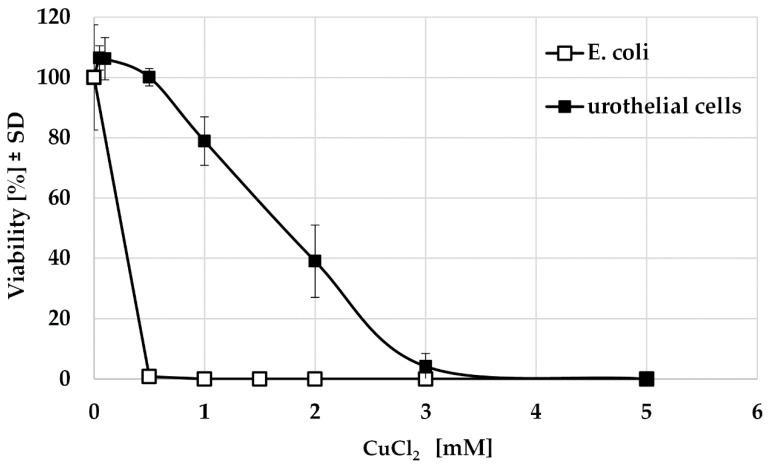
Viability of E. coli and urothelial cells HUC-1 after 24 h stimulation with CuCl_2_ (*n* = 3).

**Figure 4 polymers-14-03324-f004:**
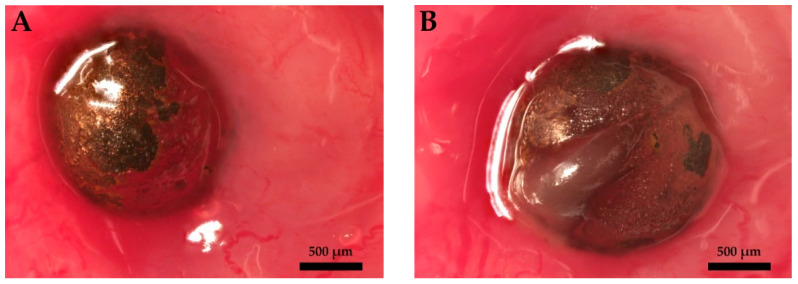
Rat urinary bladder implants. Sampling on postoperative day 7. Large protein layers were observed on the surfaces of the implants. The copper coating remained on the entire surface of the coated glass bead. (**A**): Borosilikate glass, 2.0 mm bead, copper coating, 1.0 µm, (**B**): copper grade 200, 2.0 mm solid copper bead, O.F.H.C copper, 99.9%, oxygen-free.

**Figure 5 polymers-14-03324-f005:**
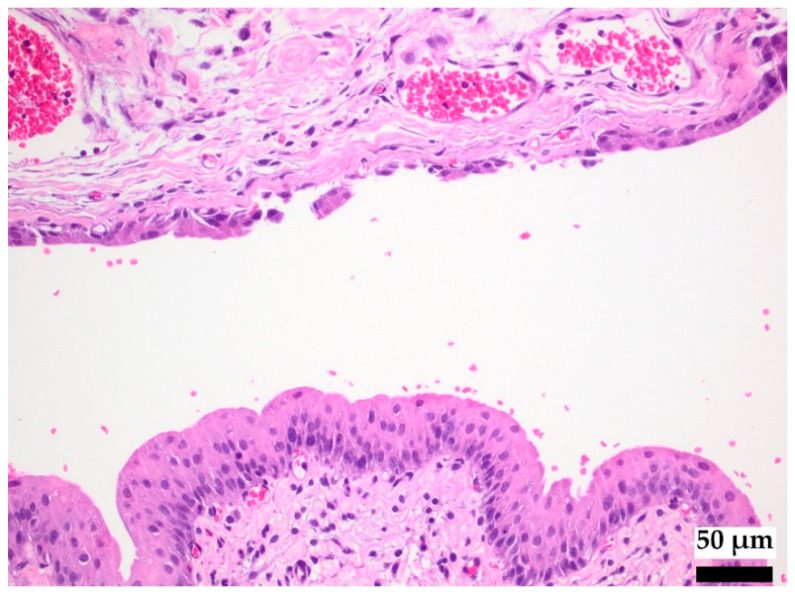
Protamine-damaged urothelium of the urinary bladder in the rat, shown by HE staining.

**Figure 6 polymers-14-03324-f006:**
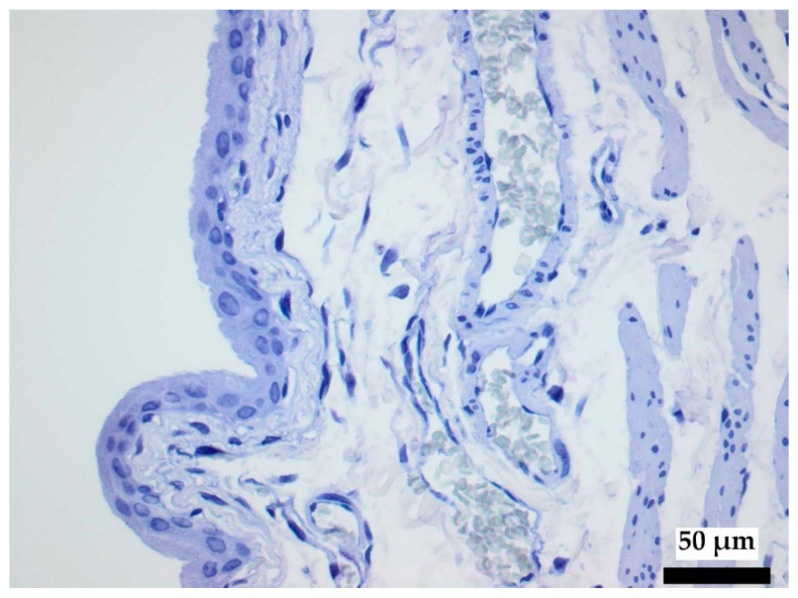
Rhodanine staining of urinary bladder tissue, with no evidence of copper.

**Figure 7 polymers-14-03324-f007:**
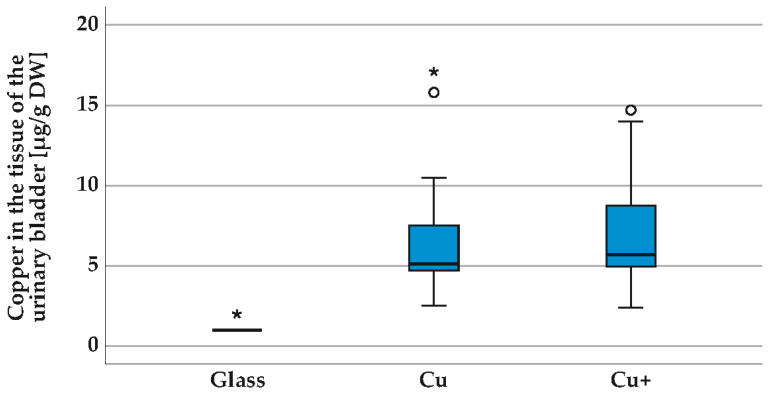
Copper in rat bladder tissue after sampling on day 21, (in µg/g dry weight). Groups: glass beads (glass), copper-coated glass beads (Cu) and copper beads (Cu+), groups with *n* = 18. Implant diameter 2.0 mm. The copper values in the bladder tissue for the group glass beads (glass) are within the detection limit of 1 µg/g dry weight. All measured copper values in the tissue are in the non-toxic range. Flame atomic absorption spectrometry (F-AAS, AAnalyst400, Perkin Elmer, Rodgau, Germany), median with scatter in box and whisker plot, Kruskal–Wallis test, groups glass-Cu and glass-Cu+ *p* < 0.001, group Cu-Cu+ *p* > 0.05, ° and * are outliers.

**Figure 8 polymers-14-03324-f008:**
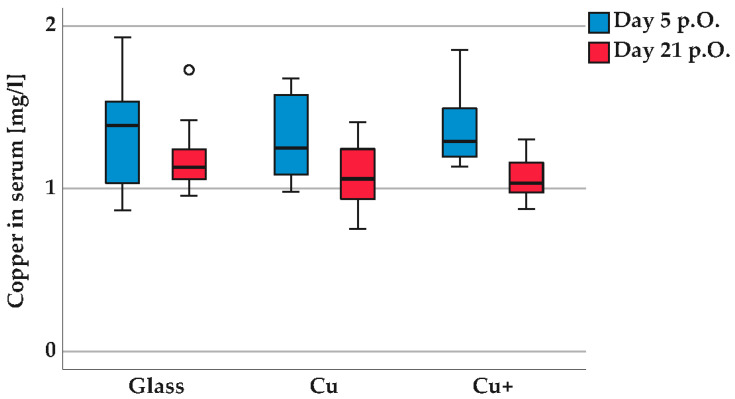
Copper in rat serum on days 5 and 21 after implantation (in mg/L). Groups: glass beads (glass), coated glass beads (Cu), copper beads (Cu+). Mass spectrometry with inductively coupled plasma (ICP-MS). Median with scatter in box and whisker plot. There are no statistically significant differences between the experimental groups, paired sample *t*-test: *p* > 0.05 (groups glass, Cu, Cu+ day 5 p.O. and day 21 p.O.), Kruskal–Wallis test: *p* > 0.05, (groups glass-Cu, glass Cu+, Cu-Cu+, p day 5 p.O. and day 21 p.O.).

**Figure 9 polymers-14-03324-f009:**
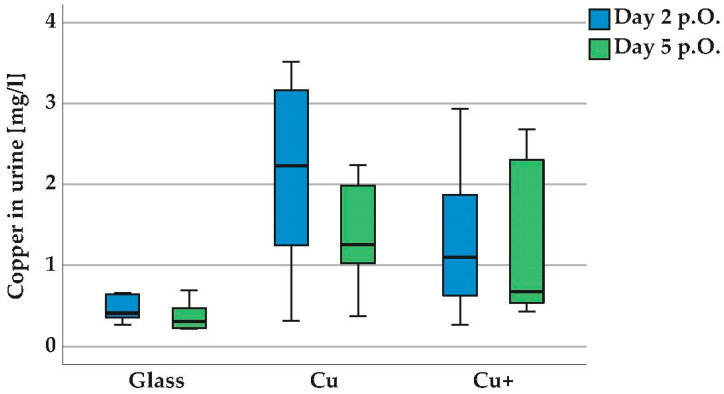
Copper in rat urine on days 2 and 5 after implantation (in mg/L). Groups glass beads (glass), coated glass beads (Cu) and copper beads (Cu+). Inductively coupled plasma mass spectrometry (ICP-MS). The concentrations of copper in the urine of the rats are increased in all groups. Median with scatter in the box and whisker plot. There are statistically significant differences between the group Cu, postoperative day 2 and day 5, *p* < 0.05, paired sample t-test, also between the groups glass-Cu, postoperative day 2 and day 5, *p* < 0.05, Kruskal–Wallis test.

**Figure 10 polymers-14-03324-f010:**
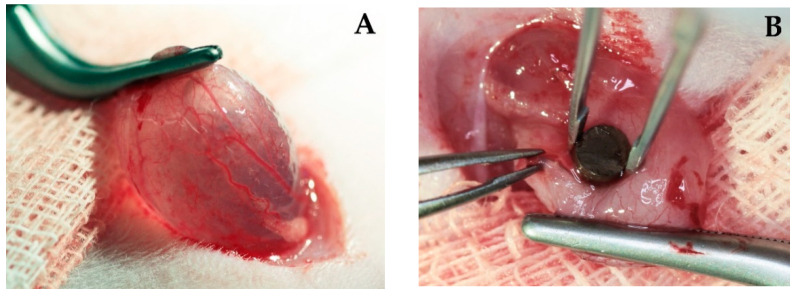
(**A**) rat urinary bladder, (**B**) implantation of a sample: Elastollan + barium sulphate + amorphous carbon layer, in combination with a-C:H/Cu-mulitilayer coating.

**Figure 11 polymers-14-03324-f011:**
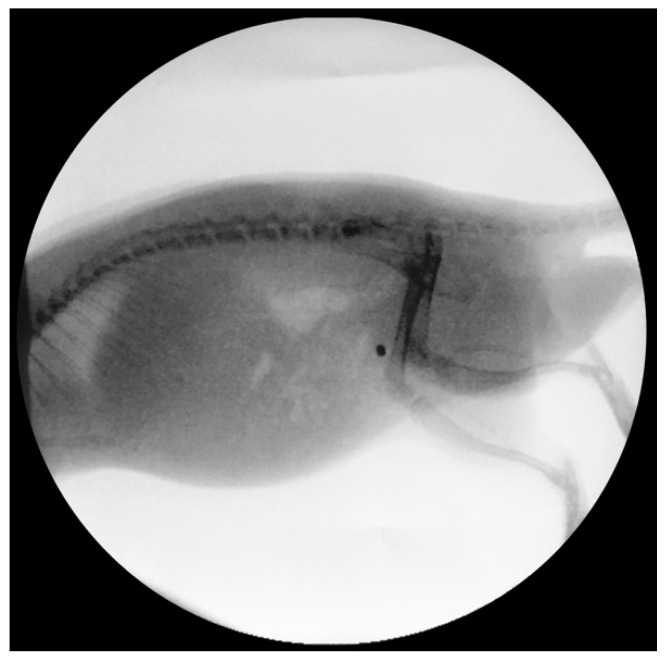
X-ray image of a rat, intravesical implant (2 × 2) mm, Elastollan + barium sulphate + a-C:H/Cu-mulitilayer coating. The selected concentration of 25% barium sulphate in the base material shows sufficient X-ray contrast. X-ray C-arm, Ziehm Vista, Zoom 2, 11 cm, Ziehm Imaging GmbH, Berlin, Germany.

**Figure 12 polymers-14-03324-f012:**
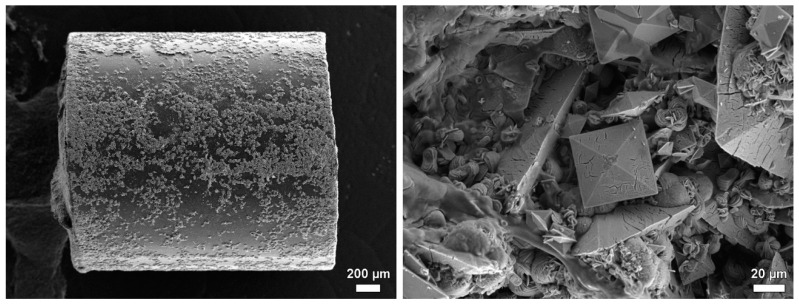
SEM of an encrusted sample surface, Elastollan + barium sulphate + a-C:H/Cu-mulitilayer coating, indwelling time 21 days. FE-SEM Merlin VP compact (Zeiss, Germany).

**Figure 13 polymers-14-03324-f013:**
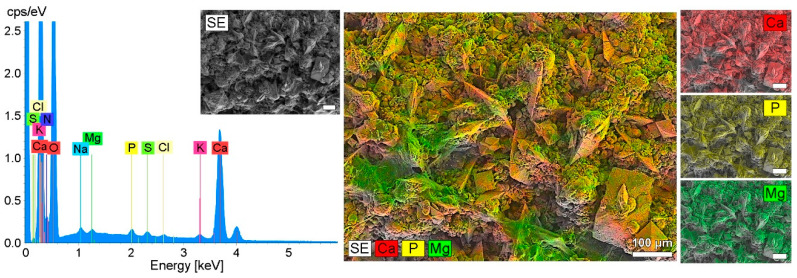
Encrustations on an implant from the urinary bladder of a rat, indwelling time 21 days. Elastollan + barium sulphate + a-C:H/Cu-mulitilayer coating. EDX line spectrum with SEM of the encrustation and element distribution of Ca, P and Mg encrustation by EDX mapping and merged image. FE-SEM Merlin VP compact (Zeiss, Germany) with EDX detector XFlash 6/30 (Bruker, Bremen, Germany).

**Figure 14 polymers-14-03324-f014:**
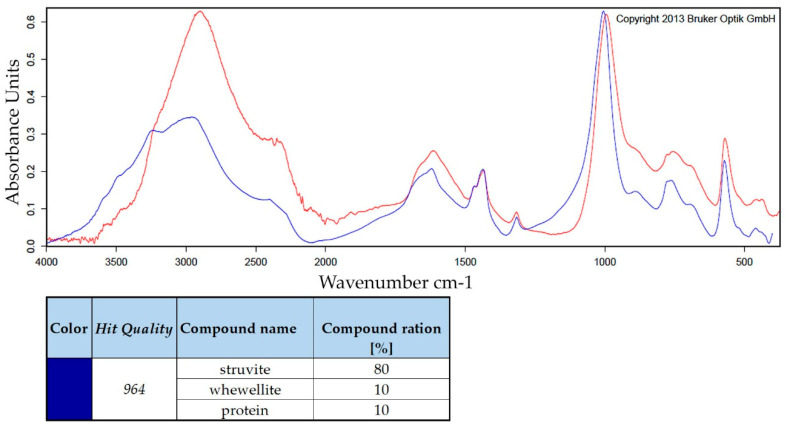
FTIR analysis of encrustation on an implant from the urinary bladder of a rat. For the Elastollan + barium sulphate + a-C:H/Cu-mulitilayer coating, the absorption spectrum (red) was measured and the absorption spectrum of the OPUS™ reference library had the best match (blue). Encrustation consists of struvite, whewellite and protein (80/10/10)%.

**Figure 15 polymers-14-03324-f015:**
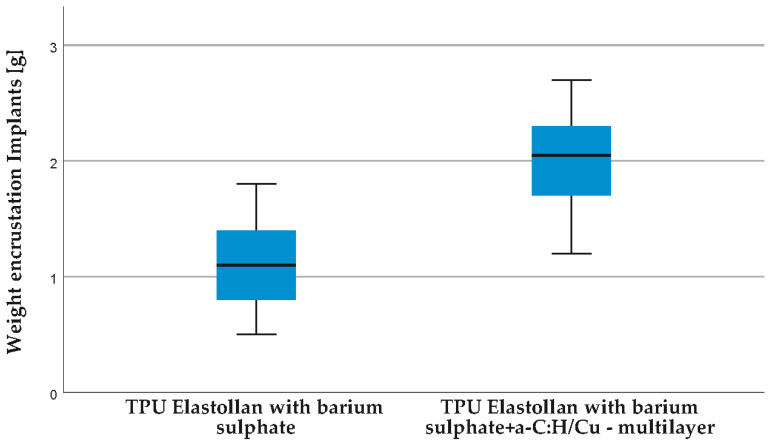
Weights of encrusted implants groups A and B. Indwelling time 21 days. Group A, Elastollan + barium sulphate, median 1.100 g. Group B, Elastollan + barium sulphate + a-C:H/Cu-mulitilayer coating, median 2.050 g. Median is shown with scatter in box and whisker plot. Sample *t*-test is used (*n* = 18, groups *p* < 0.001, effect size: 0.4 (Hedges’ correction)).

**Figure 16 polymers-14-03324-f016:**
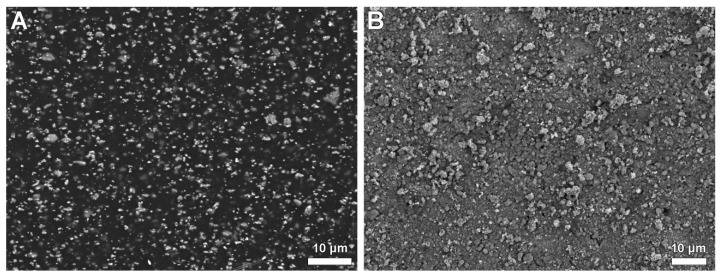
SEM of implants for urinary bladder rat before implantation. (**A**) group A, Elastollan + barium sulphate. (**B**) group B, Elastollan + barium sulphate + a-C:H/Cu-mulitilayer coating. FE-SEM Merlin VP compact (Zeiss, Germany), magnification (**A**): 1.4 × 10^3^, bar 10 µm, (**B**): 1.12 × 10^3^, bar 10 µm.

**Table 1 polymers-14-03324-t001:** Process parameters of the PECVD/PVD process for a-C:H/Cu-multilayer deposition.

No.	Process	Gasflow	Pressure Mbar	RF Excitation Power/W	Bias Voltage/V	Time/min	Thickness/nm
1	PECVD a-C:H	7 sccm Ar + 50 sccm C_2_H_2_	2.3 × 10^−2^	250	230	10	200
2	PVD magnetron sputtering Cu	50 sccm Ar	3.8 × 10^−2^	150	130	10	200

## Data Availability

Not applicable.

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
