# Peer review of "Interactive Effects of Copper-Doped Urological Implants with Tissue in the Urinary Tract for the Inhibition of Cell Adhesion and Encrustation in the Animal Model Rat"

_polymers, 2022, doi:10.3390/polym14163324_

Round 1

Reviewer 1 Report

In recent years, with the increasing number of patients with urinary system diseases, the corresponding biomedical devices are playing a more and more important role in the diagnosis and treatment of diseases and the improvement of patients' quality of life. As a relevant patient, I feel the same about the requirements of the device, so I recommend that this paper be published after minor revision

1.     The Graphic Abstract could be revised as Figure 1. In fact, I saw Graphic Abstract on Polymer for the first time.

2.     Please modify the paper in strict accordance with the template format of the journal. The current format is not the standard Polymer format.

3.     In the figure legends, some sentences are capitalized and some are lowercase. Please unify them.

4.     In Figure 1, I can’t see why n=3? Where’s the SD value? The data should be presented as (mean±SD, n=3)

5.     In Figure 2, the picture looks as if it has been pulled disproportionately, because the text seems to be out of shape. Please give an isometric picture or original drawing.

6.     The initial letter of the vertical coordinate label of some figures is not capitalized, please check

7.     Figure 3, what’s the difference between the 2 images. Please mark them in the figure legend. The scalebar is not clear, please enlarge them.

8.     Figure 4, The scalebar is not clear, please enlarge them.

9.     Figure 5 legend, what is “Rhodanine”? Do you mean “Rhodamine”? The scalebar is missing.

10.   Figure 6, the words in the figure is too small to be seen clearly.

11.   Figure 7, the words in the figure is too small to be seen clearly.

12.   Figure 8, the words in the figure is too small to be seen clearly.

13.   Figure 13, the words in the table is too small to be seen clearly.

14.   In the Introduction part, a recent reference about metals ions on human cells is suggested (Y. Hou et al., The increased ratio of Mg2+/Ca2+ from degrading magnesium alloys directs macrophage fate for functionalized growth of endothelial cells, Smart Materials in Medicine 3 (2022) 188-198.)

Author Response

Point-by-Point Response to Reviewers`Comments on Manuscript ID: polymers-1795714

Title: Interactive effects of copper-doped urological implants with tissue in the urinary tract for the inhibition of cell adhesion and encrustation in the animal model rat.

Firstly, we would like to express our gratitude to the Editors and Reviewers who spent time to review and evaluate our work for publication in polymers.

Secondly, this is a point-by-point report for response to comments from the Reviewers. Each comment has been presented individually and followed by a response statement.

Responses to Comments from Reviewer #1:

Comment:

  1. The Graphic Abstract could be revised as Figure 1. In fact, I saw Graphic Abstract on Polymer for the first time.

Response: The Graphic Abstract meets “Instructions for Authors”.

https://www.mdpi.com/journal/polymers/instructions

If the editor agrees, the Graphic Abstract should be included as Figure 1 with appropriate legend. It is a combination of real microscopic images supplemented by an artificial representation of all conditions for the interaction between stent and urothelial tissue.

Comment:

  1. Please modify the paper in strict accordance with the template format of the journal. The current format is not the standard Polymer format.

Response: The references were processed using the Endnote literature database. The style of the journal Polymers MDPI was used for the manuscript (MDPI Chicago) Font Palatino Linotype 10, for images Linotype 12 Bold. It was agreed with the publisher that the manuscript with the references would be submitted in Endnote format.

Comment:

  1. In the figure legends, some sentences are capitalized and some are lowercase. Please unify them.

Response: The figure legends have been revised.

Comment:

  1. In Figure 1, I can’t see why n=3? Where’s the SD value? The data should be presented as (mean±SD, n=3)

Response: The figure has been revised.

Comment:

  1. In Figure 2, the picture looks as if it has been pulled disproportionately, because the text seems to be out of shape. Please give an isometric picture or original drawing.

Response: The figure has been revised.

Comment:

  1. The initial letter of the vertical coordinate label of some figures is not capitalized, please check

Response: The figure has been revised.

Comment:

  1. Figure 3, what’s the difference between the 2 images. Please mark them in the figure legend. The scalebar is not clear, please enlarge them.

Response: The figure 3 has been revised. Figure 3 (A) (new figure 4A and B) shows a glass bead with a quantitatively defined copper coating, (B) a solid copper bead, group Cu+ for maximum copper release. After seven days of indwelling time, residues of the copper coating were detected on the borosilicate glass. With reference to in vitro experiments with the same implants in synthetic urine, no residual copper coating was expected. Presumably, protein coatings delayed copper release. Copper is characterised as a labile metal. Due to the release and the rapidly changing surface of the solid copper bead, no protein deposits were expected.

Comment:

  1. Figure 4, The scalebar is not clear, please enlarge them.

Response: The figure has been revised.

Comment:

  1. Figure 5 legend, what is “Rhodanine”? Do you mean “Rhodamine”? The scalebar is missing.

Response: The figure has been revised. Rhodanine: (5-p-dimethylaminobenzylidene-rhodanine) binds to both the free copper moiety and copper-associated proteins, forming a reddish-brown colour complex (Wilson's Disease Stain, Bio Optica Milano S.p.A., Milano, Italy). A blue nuclear stain with haemalaun was used for contrast.

Comment:

  1. Figure 6, the words in the figure is too small to be seen clearly.
  2. Figure 7, the words in the figure is too small to be seen clearly.
  3. Figure 8, the words in the figure is too small to be seen clearly.
  4. Figure 13, the words in the table is too small to be seen clearly.

Response: All figures has been revised ( Linotype 12 Bold).

Comment:

  1. In the Introduction part, a recent reference about metals ions on human cells is suggested (Y. Hou et al., The increased ratio of Mg2+/Ca2+ from degrading magnesium alloys directs macrophage fate for functionalized growth of endothelial cells, Smart Materials in Medicine 3 (2022) 188-198.)

Response: The reference has been included, it is a valuable addition for the problems of initial reactions between stents and the microenvironment in the blood vessels. Many thanks.

Reviewer 2 Report

The scientific paper "Interactive effects of copper-doped urological implants with tissue intheurinary tract for the inhibition of cell adhesion and encrustation in the animal model rat” aimed to investigate the interaction of a DLC coating (diamond like carbon, alsocalled amorphous hydrogenated carbon with a-C:H) in combination with copper doping (a- C:H:Cu) with the tissue of the rat urinary bladder and the encrustation of the implants. I can make the following considerations:

1)       The authors of the manuscript must adjust the references according to the rules of the Polymers MDPI journal.

2)       In the abstract, the results section is very poor. Please complete.

3)       In the introduction, the first paragraph does not contain any bibliography. Is that correct?

4)       In the introduction, do Elastollan, Styroflex and Greenflex products contain a trademark? Which manufacturer, city, country? Please complete.

5)       The results and discussion section is poorly organized. I suggest that its construction obeys, in each analysis, the positioning of the results obtained and, in the sequence, the discussion with the literature.

6)       The figures from the in vivo study, as well as the histological images, are not of good quality.

Author Response

Point-by-Point Response to Reviewers`Comments on Manuscript ID: polymers-1795714

Title: Interactive effects of copper-doped urological implants with tissue in the urinary tract for the inhibition of cell adhesion and encrustation in the animal model rat.

Firstly, we would like to express our gratitude to the Editors and Reviewers who spent time to review and evaluate our work for publication in polymers.

Secondly, this is a point-by-point report for response to comments from the Reviewers. Each comment has been presented individually and followed by a response statement.

Responses to Comments from Reviewer #2:

The scientific paper "Interactive effects of copper-doped urological implants with tissue intheurinary tract for the inhibition of cell adhesion and encrustation in the animal model rat” aimed to investigate the interaction of a DLC coating (diamond like carbon, also called amorphous hydrogenated carbon with a-C:H) in combination with copper doping (a- C:H:Cu) with the tissue of the rat urinary bladder and the encrustation of the implants. I can make the following considerations:

Comment:

  • The authors of the manuscript must adjust the references according to the rules of the Polymers MDPI journal.

Response: The references were processed using the Endnote literature database. The style of the journal Polymers MDPI was used for the manuscript (MDPI Chicago) Font Palatino Linotype 10, for images Linotype 12 Bold. It was agreed with the publisher that the manuscript with the references would be submitted in Endnote format.

Comment:

  • In the abstract, the results section is very poor. Please complete.

Response: The abstract was edited.

Comment:

  • In the introduction, the first paragraph does not contain any bibliography. Is that correct?

Response: A reference was added to the introduction, the first paragraph. The statements correspond to the current state of knowledge of the urological professional societies.

Comment:

  • In the introduction, do Elastollan, Styroflex and Greenflex products contain a trademark? Which manufacturer, city, country? Please complete.

Response: The additional data of the products tested by the authors have been inserted.

Comment:

  • The results and discussion section is poorly organized. I suggest that its construction obeys, in each analysis, the positioning of the results obtained and, in the sequence, the discussion with the literature.

Response: The entire structure of the manuscript was edited.

Comment:

  • The figures from the in vivo study, as well as the histological images, are not of good quality.

Response: All images have been edited. The final quality meets the requirements of the Polymers MDPI journal. In the Word file (PDF-file), the smaller format jpeg has been used to reduce the file size.

Reviewer 3 Report

Comments

The authors reported the biocompatibility, release of copper ions and the accumulation of encrustations with the insertion of copper-doped urological implants into an animal model of rat. This study was well-designed and deeply investigated. However, despite the antimicrobial activity and anti-adhesion property of the copper-coated implants, do the authors consider the accumulation effect of copper ions? Though the author demonstrated that the copper ions with concentration of 0.5-1.0 mM could effectively kill bacteria without affecting the urothelium. However, as one kind of heavy metal ions, coppers ions could be accumulated in our body and leads to the increase of the local concentration in different organs, such as liver and spleen. How do the authors consider this question? Besides, there are also many issues should be addressed.

1.      There is usually one paragraph in the abstract section. Please combine the four paragraphs into one paragraph.

2.      At the first place of the appearance of abbreviations, the full name should be provided. For instance, it is hard to understand the ‘a-C:H:Cu coating’ in the abstract. Besides, once the abbreviations have been defined, the full name should be avoided in the following text. The use of abbreviations and full names in this article is very confusing.

3.      Materials and their providers, as well as the preparation procedures of copper-doped urological implants should be provided in the materials and methods section in order to repeat the study.

4.      At the results and discussion section, there are lot of content about the summary and discussion of previous literatures, which should be moved to the introduction section. It should be avoided to glance through the previous studies in the results and discussion section. On the contrary, results and figures and corresponding analysis should be the focus of the results and discussion section.

5.      What is the difference between ‘a:C:H:Cu’ ‘a-C:H:Cu’ and ‘a:CH:Cu’? Or they are the same, which is confused to me.

6.      In Figure 7, the concentration of copper in serum on day 5 is higher than that on day 21 overall, why?

7.      There is no need to define the abbreviations repeatedly, especially in pages 12 and 13.

8.      What is the table meaning in Figure 13?

9.      XRD analysis may be more convinced to determine the existence and structures of crystals instead of FTIR.

10.  The unity of the vertical axis of Figure 14 is mg, while the unity in the caption is g. Besides, the following explanation also cannot match the Figure 14.

11.  Figure 15 is the SEM of implants for urinary bladder rat before implantation. Then where is the the encrustation from?

12.  There are lots of references are cited in the discussion section to prove the statement of the authors. I would suggest the authors to offer more evidence rather than citing references to prove the statements or conclusions since this is an original research article.

13.  In the conclusion section, the authors stated that the animal model is pig. but actually, the animal model used in this study is rat.

Author Response

Point-by-Point Response to Reviewers`Comments on Manuscript ID: polymers-1795714

Title: Interactive effects of copper-doped urological implants with tissue in the urinary tract for the inhibition of cell adhesion and encrustation in the animal model rat.

Firstly, we would like to express our gratitude to the Editors and Reviewers who spent time to review and evaluate our work for publication in polymers.

Secondly, this is a point-by-point report for response to comments from the Reviewers. Each comment has been presented individually and followed by a response statement.

Responses to Comments from Reviewer #3:

Comments

The authors reported the biocompatibility, release of copper ions and the accumulation of encrustations with the insertion of copper-doped urological implants into an animal model of rat. This study was well-designed and deeply investigated. However, despite the antimicrobial activity and anti-adhesion property of the copper-coated implants, do the authors consider the accumulation effect of copper ions? Though the author demonstrated that the copper ions with concentration of 0.5-1.0 mM could effectively kill bacteria without affecting the urothelium. However, as one kind of heavy metal ions, coppers ions could be accumulated in our body and leads to the increase of the local concentration in different organs, such as liver and spleen. How do the authors consider this question? Besides, there are also many issues should be addressed.

Response: see comment 4

Comment:

  1. There is usually one paragraph in the abstract section. Please combine the four paragraphs into one paragraph.

Response: has been edited, thanks for the recommendation.

Comment:

  1. At the first place of the appearance of abbreviations, the full name should be provided. For instance, it is hard to understand the ‘a-C:H/Cu coating’ in the abstract. Besides, once the abbreviations have been defined, the full name should be avoided in the following text. The use of abbreviations and full names in this article is very confusing.

Response: has been edited, thanks for the recommendation.

See in the manuscript:

“In the present in vivo studies, the interaction of amorphous hydrogenated carbon composite coatings (a-C:H), also called diamond like carbon (DLC), in combination with copper doping (a-C:H/Cu - multilayer) with the tissue of the rat urinary bladder and the encrustation of the implants were investigated.

Comment:

  1. Materials and their providers, as well as the preparation procedures of copper-doped urological implants should be provided in the materials and methods section in order to repeat the study.

Response:

Supplement: 2.1 Preparation of implants    

2.1.1.   Implants with copper coating beads

2.1.2.   Impants with a-C:H/Cu - mulitilayer coating

Comment:

  1. At the results and discussion section, there are lot of content about the summary and discussion of previous literatures, which should be moved to the introduction section. It should be avoided to glance through the previous studies in the results and discussion section. On the contrary, results and figures and corresponding analysis should be the focus of the results and discussion section.

Response: In the introduction, the authors describe the main task and the state of the art to describe the intended purpose of the studies. We fully agree with the reviewer that "in addition, many questions should still be addressed". A detailed account of the results should be given in the discussion. 

An important point here is the antibacterial effect and the toxic effect of copper. A central concern, analogous to the reviewer's introduction, is the question of the accumulation of copper in other organs:

“However, as one kind of heavy metal ions, coppers ions could be accumulated in our body and leads to the increase of the local concentration in different organs, such as liver and spleen.”

This issue has not been sufficiently discussed in the literature so far. The areas of the antibacterial effect of copper on bacteria at the cellular level (both as metal ions and as direct contact toxicity) and, on the other hand, the disease value, the pathological, toxic effect on the organism must be considered. In the introduction concerning the point Results and Discussion, it was first shown that genetic copper storage disease (one in thousand people are affected) can be regarded as an extreme situation for copper toxicity. The pathological processes as well as the diagnostic and therapeutic steps are well researched and summarised in the general treatment guidelines for Wilson's disease, reference [34]. The target organ of copper toxicity is usually the liver. Mentioned in the manuscript, page8:

“In humans, toxic copper accumulation and the associated dysfunction particularly affect the liver, the central nervous system, the eyes and the kidneys. / …… Excess copper accumulates as free copper and is excreted via the liver, kidneys and intestines.” Reference [39]

However, for copper-releasing stents, the site of entry is the adjacent tissue of the urinary bladder.  In the present study, this point was particularly evaluated because of the damage to the bladder tissue at the point of contact. References for copper in bladder tissue are not known. Only the content of copper in serum and urine as a result of copper metabolism is known.

Comment:

  1. What is the difference between ‘a:C:H:Cu’ ‘a-C:H:Cu’ and ‘a:CH:Cu’? Or they are the same, which is confused to me.

Response: This is a spelling mistake, the correct is a-C:H/Cu - mulitilayer, thanks for the recommendation.

Comment:

  1. In Figure 7, the concentration of copper in serum on day 5 is higher than that on day 21 overall, why?

Response: (new figure 8) The aim was to prove that the Cu levels in the blood are in the non-toxic range. It is true that the median Cu level on day 5 is higher than on day 21. One could assume that a small amount of copper enters the blood vessels through the damaged urothelium. Statistically, this could not be proven. Furthermore, the GLASS reference group also shows elevated values.

Comment:

  1. There is no need to define the abbreviations repeatedly, especially in pages 12 and 13.

Response: has been edited, thanks for the recommendation. In the manuscript, only the explained abbreviations are used after the definition.

Comment:

  1. What is the table meaning in Figure 13?

Response: Figures 13 (new figure 14): Measured absorption spectrum (red), absorption spectrum of the OPUS™ reference library with the best match (blue). The original table of the analysis software has been changed.

Comment:

  1. XRD analysis may be more convinced to determine the existence and structures of crystals instead of FTIR.

Response: XRD analysis and FTIR complement each other in the evaluation of incrustations. XRD analysis can only detect chemical elements. Copper was not detected. With mapping, the qualitative distribution can be assessed very well. Difficult is the exact analysis of the composition of the encrustation, which usually consists of a mixture of substances. The exact analysis of the substance provides information about the causes of the encrustation, especially about the chemical processes. For this reason, urological societies recommend urinary stone analysis with FTIR.

Comment:

  1. The unity of the vertical axis of Figure 14 is mg, while the unity in the caption is g. Besides, the following explanation also cannot match the Figure 14.

Response: The unit of the vertical axis of Figure 14 (new Figure 15) is wrong, correct is g.

Figure 14 shows the weights of the encrusted implants groups A and B. Median with scatter in box and whisker plot. Indwelling time 21 days.  In group B, implants with a-C:H/Cu a significantly higher tendency to encrustation was observed.

Comment:

  1. Figure 15 is the SEM of implants for urinary bladder rat before implantation. Then where is the the encrustation from?

Response: Figure 15 (B) (new Figure 16) shows an increased roughness due to the copper coating compared to the base material figure (A) and should support the assumption that crystals and also bacteria from the urine could be separated better here.

Comment:

  1. There are lots of references are cited in the discussion section to prove the statement of the authors. I would suggest the authors to offer more evidence rather than citing references to prove the statements or conclusions since this is an original research article.

Response: With regard to comment 4, the authors would like to note that the interdisciplinary nature of the concept of copper toxicity requires that recognised references can support the own findings and provide further information.

Furthermore, with reference to comment 13, it should be mentioned that the present studies are the preparation for a final in vivo study as well as the preparation for a clinical study.

The summary has been edited.

Comment:

  1. In the conclusion section, the authors stated that the animal model is pig. but actually, the animal model used in this study is rat.

Response: In the present study the animal model is a rat. The limitations of this animal model were mentioned in the manuscript, page 14. For this reason, it is pointed out that further studies and the verification of the results with the final implants (prototype of a newly developed ureteral stent based on the basic material elastollan with a-C:H/Cu - multilayer coating) should be carried out with the animal model pig. The urinary tract of the pig and also the urine have the greatest similarity to the urinary tract of humans. However, keeping the animals and conducting the study is much more complex.

Round 2

Reviewer 2 Report

No comments

Reviewer 3 Report

All of my questions have been addressed. Therefore, I think that the manuscript can be published in Polymers in its current form.